# TNFα and Immune Checkpoint Inhibition: Friend or Foe for Lung Cancer?

**DOI:** 10.3390/ijms22168691

**Published:** 2021-08-13

**Authors:** Thomas Benoot, Elisa Piccioni, Kirsten De Ridder, Cleo Goyvaerts

**Affiliations:** Laboratory for Molecular and Cellular Therapy, Department Biomedical Science, Vrije Universiteit Brussel (VUB), Laarbeeklaan 103, 1090 Brussels, Belgium; thomas.benoot@vub.be (T.B.); piccioni.1750442@studenti.uniroma1.it (E.P.); kirsten.de.ridder@vub.be (K.D.R.)

**Keywords:** TNFα, TNFR1, TNFR2, lung cancer, immunotherapy, immune checkpoint inhibition (Min. 5–Max. 8)

## Abstract

Tumor necrosis factor-alpha (TNFα) can bind two distinct receptors (TNFR1/2). The transmembrane form (tmTNFα) preferentially binds to TNFR2. Upon tmTNFα cleavage by the TNF-alpha-converting enzyme (TACE), its soluble (sTNFα) form is released with higher affinity for TNFR1. This assortment empowers TNFα with a plethora of opposing roles in the processes of tumor cell survival (and apoptosis) and anti-tumor immune stimulation (and suppression), in addition to angiogenesis and metastases. Its functions and biomarker potential to predict cancer progression and response to immunotherapy are reviewed here, with a focus on lung cancer. By mining existing sequencing data, we further demonstrate that the expression levels of *TNF* and *TACE* are significantly decreased in lung adenocarcinoma patients, while the *TNFR1*/*TNFR2* balance are increased. We conclude that the biomarker potential of TNFα alone will most likely not provide conclusive findings, but that TACE could have a key role along with the delicate balance of sTNFα/tmTNFα as well as TNFR1/TNFR2, hence stressing the importance of more research into the potential of rationalized treatments that combine TNFα pathway modulators with immunotherapy for lung cancer patients.

## 1. The Pleiotropic Immunological Biology of TNFα

About 60 years ago, it was reported that bacterial endotoxin administration to mice resulted in the release of a serological protein with necrotic anti-tumor activity at high concentrations. Due to the latter characteristic, this protein was termed tumor necrosis factor (TNF) [1] and considered a breakthrough for cancer therapy. Today, the TNF superfamily consists of 19 members and 29 TNF receptors [2]. Within this family, functional TNFα is represented by a trimer of 17.35 kDa monomers, folded into a rigid bell-shaped “jelly roll” composed of antiparallel filaments [3]. It exists in two forms: a transmembrane form (tmTNFα) next to a soluble (sTNFα) form. The latter one is cleaved from tmTNFα by the metalloproteinase TNF-alpha-converting enzyme (TACE). Upon X-ray crystallography analysis, TNFα was demonstrated to bind to TNF receptors 1 and 2 (TNFR1 and TNFR2), represented by a 55 and 75 kDa type I and type II transmembrane protein, respectively [4,5]. The structural pleiotropism of TNFα and its receptors endows it with a multifaceted role linked to both anti- and pro-inflammatory assets as well as apoptotic assets. The complexity of these findings tempered the original enthusiasm for TNFα as a breakthrough molecule for cancer therapy.

### 1.1. The Pro-Inflammatory Character of sTNFα versus tmTNFα

Soluble TNFα is mainly secreted by activated macrophages [6] and to a lesser extent, by T lymphocytes, natural killer (NK) cells, neutrophils, endothelial and cardiac muscle cells, fibroblasts, and osteoclasts [7,8]. By comparison, tmTNFα is expressed constitutively on the surface of a broad range of immune cells such as alveolar and non-alveolar macrophages [9], monocytes [10], lymphocytes [11], dendritic cells (DCs), and NK cells [12]. In addition, its expression has been reported on non-immune cells such as adipocytes [13] and tumor cells [14].

In general, sTNFα is rapidly released upon trauma or infection, as it is bestowed with a determining role in immunoregulatory processes such as immune ontogeny, inflammation, and apoptosis [7]. As a soluble pro-inflammatory cytokine, it primarily acts at sites remote from the TNFα-producing cells to support the production of downstream pro-inflammatory cytokines along with the recruitment, activation, and regulation of inflammatory cells such as macrophages. To illustrate, when macrophages are activated by Toll-like receptors, they secrete sTNFα, which subsequently regulates macrophage differentiation in an autocrine fashion [15]. Hence, TNFα neutralizing antibodies have been shown to reduce the production of several pro-inflammatory cytokines and growth factors such as interleukin-1 (IL-1) and granulocyte-macrophage colony-stimulating factor (GM-CSF) [16]. Of note, sTNFα has an intrinsic pleiotropic activity as it is also involved in anti-inflammatory responses that aim to restore homeostasis [17].

In comparison, tmTNFα is an important mediator of immune-cell crosstalk. To illustrate, when DCs express tmTNFα, the latter can interact with TNFR2 on NK cells, resulting in increased NK cell proliferation and cytotoxic activity [18]. Moreover, tmTNFα can act in a dual manner upon its interaction with TNFR: either as a ligand or as a receptor through an “outside-to-inside” signaling pathway known as “reverse signaling”. There is evidence that, as a ligand, it is mainly involved in host defense mechanisms against infections [19], while the receptor-like form appears to have a role in modulating immune cell activation. To elaborate, tmTNFα expressed by activated T cells can bind to TNFR2, expressed by monocytes, leading to monocyte activation with subsequent secretion of interleukin-10 (IL-10) and sTNFα. Additionally, endogenous IL-10 was shown to downregulate T cell contact-mediated sTNFα production by monocytes, suggestive for an autoregulatory loop involving both sTNFα and tmTNFα [19,20,21]. What is more, T cells can also activate monocytes by expressing TNFR2 that can trigger reverse signaling via tmTNFα, expressed by the monocytes. Hence, both TNFα isoforms are significantly involved in the regulation of the inflammatory response [22]. 

### 1.2. Two Distinct Receptors Fine-Tune TNFα’s Biological Effects

TNFR1, also known as tumor necrosis factor receptor superfamily, member 1A (TNFRSF1A) or CD120a, is expressed on almost all host cells including various tumor cell types [23,24,25,26] and tumor-associated endothelial cells [27]. In contrast, TNFR2 (TNFRSF1B or CD120b) is predominantly located on the surface of immune cells, such as NK cells, macrophages [28], regulatory T cells (Tregs), suppressive myeloid cells [29], and endothelial cells [30]. As indicated by "+" in Figure 1, sTNFα shows a significantly higher affinity for TNFR1 than tmTNFα, while the opposite holds true for TNFR2 [31,32]. 

In terms of biological effects, high affinity binding to TNFR1 (and not TNFR2) has been demonstrated to result in DC activation and subsequent stimulation of antigen specific CD8^+^ T cells [33]. That TNFR1-knockout (ko) mice show resistance to colitis development [34] further confirms its involvement in the differentiation of inflammatory T cells. In comparison, activation of TNFR2 in lymphoid cells leads to inflammatory responses as well as T cell activation, thymocyte proliferation, GM-CSF production [7], and NK cell-mediated IFN-γ production [35]. Yet, its main role seems to be linked to restoration of homeostasis and promotion of an immunosuppressive environment, as TNFR2 gene ko models show elevated inflammatory responses [33,34,36]. To illustrate, TNFR2 signaling has been shown to promote Treg differentiation, proliferation, and suppressive function, while TNFR1 does not affect Treg cell expansion [37,38,39]. Moreover, upon activation of both TNFRs, TNFR2 is more liable to receptor shedding than TNFR1 [40,41,42]. The shedded soluble TNFRs have been shown to serve as decoy receptors for sTNFα to control its innate immune activation threshold. Hence, more pronounced TNFR2 shedding is in line with its immune restorative function [43,44].

Next to their roles in immune cell proliferation and activation, triggered TNFR1 and TNFR2 can also induce apoptotic and necrotic cell death. In the case of TNFR1, this is the result of the interaction between the receptors’ death domain (DD) and the adapter protein TNF receptor-associated death domain (TRADD). Upon interaction of DD-TRADD, the activation of cysteine-aspartyl-specific proteases (caspases) is stimulated [45,46]. Although TNFR2 lacks a DD, it has been reported to cause activation-induced cell death (AICD), e.g., in mature CD8^+^ T cells [36]. As outlined in Figure 1, the different outcomes upon TNFR1 and 2 triggering result from their distinct downstream signaling [47]. Although both exploit the NF-κB pathway for transcriptional activation of inflammatory and anti-apoptotic genes [48], the “classical” NF-κB pathway is mainly activated upon sTNFα binding to TNFR1 [31], while the “alternative pathway” is preferentially but not exclusively primed upon TNFR2 binding [28,49,50]. Additionally, both TNFRs can regulate gene expression by activating at least two members of the MAP kinase (MAPK) family, such as P38 [51,52] and JNK [53], and this via a series of protein phosphorylations (MEKK). Notably, the TNF superfamily member lymphotoxin alpha can, in its soluble homotrimer form (LTα3), also bind to both TNFRs with an affinity profile comparable to that of sTNFα [54,55]. In terms of biological effects, sTNFα and LTα3 have been reported to show equal mitogenic stimulation capacities upon TNFR binding, yet sTNFα was shown to be more potent to mediate gene regulation and cytotoxicity.

In conclusion, the presence of two functional isoforms (soluble and transmembrane) next to two receptors with specific expression, affinity, and downstream signaling avenues collectively serves the pleiotropic job description of TNFα. As regulator of cytotoxic, pro-, and anti-inflammatory functions, it is evident that TNFα can play a critical role in the development of chronic inflammatory diseases as well as cancer, as outlined below.

## 2. TNFα Plays Opposing Roles in Cancer

Despite TNFα’s denomination, in vitro reported tumor necrosis after high TNFα concentrations appeared a phenomenon that is not so straightforwardly translated to successful cancer treatments in vivo. The latter is partly explained by TNFα’s multifunctionality as a cytotoxic but also immune modulating cytokine. As the immune system plays a complex role on the tumor microenvironmental (TME) battlefield, TNFα is used as a weapon to modulate and/or kill tumor cells, immune cells, and/or endothelial cells [57].

Previously observed direct anti-tumoral properties of TNFα in vivo are among others based on its capacity to hinder tumor-associated blood vessel formation (angiogenesis) via selective endothelial cytotoxicity and necrotic hemorrhage [58]. This selective endothelial cytotoxicity can subsequently result in hyperpermeability of tumor vessels and increased blood cell extravasation. The exact underlying mechanism remains unknown, yet van Horssen et al. [59] described the noteworthy hypothesis that tumor-residing endothelial cells are more sensitive to TNFα because they upregulate TNFR1 upon cytokine signaling by tumor cells and macrophages. However, external administration of TNFα was shown to occupy TNFR1 expressed by tumor and healthy endothelial cells without toxicity towards the latter, due to the presence of a low number of receptors on healthy cells [60]. Nevertheless, studies in mice and rats further showed a synergistic anti-tumor effect of TNFα in combination with chemotherapeutic drugs, as accumulation of the latter at the tumor site was shown to be improved [61,62,63]. Moreover, persistent high-level stimulation with TNFα in vitro inhibits endothelial cell proliferation in a dose-dependent manner [64], further supporting its anti-angiogenic effects [65,66,67]. However, Fajardo et al. hypothesized that TNFα could show both pro- and anti-angiogenic effects in vivo, depending on its local concentration. When murine TNFα was administered subcutaneously in mice, this was shown to exert opposing effects. At low doses (range: 0.01–1 ng), TNFα induced angiogenesis whereas increasing doses (1 and 5 µg) reduced this effect with complete abolishment at the highest doses [68].

While a high concentration of TNFα has been linked to hemorrhagic necrosis, it is now widely accepted that chronic exposure to TNFα is more likely to promote tumor progression. First, it has been demonstrated repeatedly that chronic inflammation, in which the innate immune system plays a leading role, can promote cancer onset as well as progression and metastasis, typifying the “never-healing-wound” character of solid cancers [69,70]. Indeed, chronic exposure to TNFα can promote cellular transformation via the induction of direct mutations and DNA damage [71] as well as via profound epigenetic changes that modulate the expression level of oncogenes and tumor suppressor genes [72]. In addition, inflammation influences epithelial-to-mesenchymal cell transition (EMT) and subsequent cancer cell invasion. Further, TNFα has been shown to affect expression of EMT-inducing transcription factors, particularly in synergy with TGFβ [73]. Moreover, TNFα associated with chronic inflammation can be held responsible for the observed phenomenon of cancer cell specific resistance to TNFα-induced cell death [74]. Specifically, chronic TNFα/TNFR1 binding increases the expression of anti-apoptotic, angiogenic, and invasive proteins via the TAK-1, MAPKs, Akt, IKK, AP-1, and NF-κB signaling pathways [28,75,76]. Notably, also the ligands and receptors of the LTα family with, among others, affinity for TNFR1 and 2, have been linked to increased carcinogenesis, as extensively reviewed elsewhere [55,77].

Even if chronic inflammation is not involved in the onset of tumor cell transformation, the immune system often becomes a co-worker during cancer progression. Today it is generally accepted that the immune system can identify and control nascent malignancies in a process called cancer immunosurveillance. In contrast, the latter can also promote tumor progression through the selection of poorly immunogenic variants and suppression of anti-tumor immunity. Together, the dual host-protective and tumor-promoting actions of immunity are referred to as cancer immunoediting and comprise three distinct phases: the elimination, equilibrium, and escape phase [78,79,80].

The elimination phase is characterized by an imbalance towards more anti-tumor immunity, installed by adaptive as well as innate immune cells. Characteristic for a potent Th1-oriented tumor-associated antigen (TAA) specific adaptive immune response is the presence of immunogenic TAAs, presented via MHC-I on the surface of tumor cells, together with Fas, TRAIL, and IFN-γ receptors and the presence of perforin, granzymes, IFN-α/β/γ, IL-1, IL-12, and TNFα within the TME. Hence, TNFα can ameliorate this phase through its involvement in activation of T cells, macrophages, and NK cells. For example, it was shown that TNFR1 signaling promotes the accumulation of anti-tumor M1 polarized tumor-associated macrophages (TAMs) by suppressing the M2-polarizing release of IL-13 from eosinophils co-recruited with inflammatory monocytes [81]. On their turn, MHC-II^high^ (M1) TAMs and granulocytes can secrete, among others, TNFα, IL-1, and IL-12 to further ameliorate a Th1-polarized anti-tumor immune profile [82].

During the equilibrium phase, anti- and pro-tumor immunity are in balance and/or immune-mediated tumor dormancy is installed [80]. It was reported that the absence of TNFR or IFN-γ promoted angiogenesis and multistage carcinogenesis in an experimentally induced pancreatic murine tumor model, suggesting that a coordinated interaction between IFN-γ and TNFα was responsible for the activation of TAA-specific cytotoxic T cells [83]. Moreover, the combination of IFN-γ and TNFα drove pancreatic tumor cells into STAT-1 and TNFR1-mediated senescence [82]. Because IFN-γ and TNFα induce senescence in numerous murine and human cancers, this may be a general mechanism for arresting cancer progression. 

In the escape phase, TNFα has several effects capable of lowering the antitumor immune response by facilitating the accumulation and/or activation of a wide range of immunosuppressive cells such as Tregs [37], regulatory B cells [84], and suppressive myeloid cells [85]. As stated before, the common factor of these immunosuppressive effects lies within TNFα binding to TNFR2. When secreted by activated CD4^+^ T-cells, TNFα has been shown to induce myelopoiesis in tumor-bearing mice. Furthermore, cells of the myeloid lineage can be recruited to become immune-suppressive regulatory myeloid cells, which decrease TAA-specific CD8^+^ T cell mediated tumor cytotoxicity [86]. Moreover, it has been reported that tmTNFα, rather than sTNFα, is able to activate immunosuppressive myeloid cells upon binding to TNFR2. To illustrate, a marked increase in immunosuppressive myeloid cell accumulation was only observed when tmTNFα was constitutively expressed on 4T1 mammary tumor cells, with subsequent promotion of NO, ROS, IL-10, and TGF-β secretion by these myeloid cells and inhibition of lymphocyte proliferation. In contrast, 4T1 overexpression of sTNFα resulted in increased lymphocyte infiltration and tumor regression [87]. TNFα also facilitates the installment of an effector T cell hostile milieu via indoleamine 2,3-dioxygenase 1 (IDO1) accumulation in the TME. To illustrate, while M2b polarized macrophage conditioned medium stimulates tumor cell proliferation and IDO1 expression in vitro, this is reduced upon TNFα neutralization [88]. As IDO1 converts tryptophan into kynurenine, tryptophan is deprived with subsequent installment of TAA-specific T cell anergy, while Treg activity, lymphangiogenesis, and neovascularization are enhanced in vivo. Finally, TNFα has been linked to cancer therapy resistance. To illustrate, in a murine triple-negative breast cancer model, resistance to the anti-angiogenic drug bevacizumab was shown to be accompanied by M2b TAM-mediated secretion of the chemokine CCL1 along with TNFα. Upon TNFα-neutralizing nanobody administration in vivo, this immunosuppressive M2b macrophage-induced resistance was overcome, supporting TNFα’s key role in resistance to bevacizumab [88].

Although both TNFα and its receptors’ precise role within the complex and ever-evolving TME are far from fully understood, substantial evidence has emerged that TNFα signaling has a paradoxical and dual role in cancer progression and the dynamic immunoediting process. While it has been held responsible for key tumor promoting features such as the maintenance of an antigen-ignorant chronic inflammatory state and immunosuppressive effects that hinder TAA-specific effector functions, its documented anti-tumor assets in vitro and in vivo hamper a clear-cut conclusion.

## 3. Specificities of TNFα in Lung Cancer Progression

Lung cancer remains the leading cause of cancer-related deaths worldwide, responsible for almost 1.8 million deaths in 2020 alone [89]. Lung cancer comprises two key types: non-small cell lung cancer (NSCLC), accounting for 80–85% of cases, and small cell lung cancer (SCLC) [90]. 

As the human respiratory tract is continuously exposed to air that can potentially contain airborne pathogens, lungs necessitate a unique, fine-tuned, and rapidly acting pulmonary immune system to maintain homeostasis [91]. When the immunohistological expression of TNFα and its receptors was evaluated in healthy human lung tissue, TNFα was shown to be particularly prominent in bronchial epithelial cells, vascular smooth muscle cells, and alveolar macrophages. In addition, TNFR1 was shown to be eminently expressed on bronchial epithelial cells and endothelial cells, while TNFR2 was expressed by nearly all cell types [92]. Therefore, it is not surprising that in the complex process of lung cancer onset, progression, and dissemination, TNFα and its receptors have been reported to play decisive roles, too. Indeed, it was recently described that *TNF* is one of the co-occurring frequently altered immune genes found within the TCGA pan-cancer lung adenocarcinoma (LUAD) dataset (*n* = 507) [93]. However, the exact effect seems to be nuanced by contradicting functions governed by their isoforms, signal strength, and downstream signaling pathways. This is illustrated by the finding that specific genetic polymorphisms of the *TNF* gene region impact lung cancer progression differently. While the 238 G>A polymorphism, found in the promoter region of TNFα, has a favorable prognostic association for NSCLC [94], the 308 G>A polymorphism constitutes an increased risk for lung cancer [95]. 

Another opposing effect is found for sTNFα and tmTNFα on lung cancer growth [96]. When murine lung tumor lines expressed tmTNFα, their engraftment resulted in the formation of small tumors with reduced tumor-associated myeloid cell infiltration, in contrast to control or sTNFα-overexpressing lines. The observed myeloid cell reduction was found to be a direct effect of tmTNFα on myeloid survival via induction of ROS-mediated cell necrosis. Furthermore, human NSCLC was shown to express varying levels of sTNFα and tmTNFα, and gene expression patterns favoring tmTNFα appeared predictive of improved lung cancer survival [96]. This is in contrast with previous findings using the mammary 4T1 model and, moreover, counterintuitive, as tmTNFα shows a higher affinity than sTNFα for the immunosuppression promoting TNFR2 [87]. The fact that TNFα could have a favorable impact on survival is further supported by a study on citronellol’s ability to induce necroptosis of human lung tumor cells in vitro and in vivo. This study identified a decisive role for TNFα in this necroptosis induction via its activation of RIP1/3 and simultaneous downregulation of caspase 3/8 activity [97]. 

In an attempt to predict the most common interactions between the different TNFα-signaling members within the lung TME, we compared the overall *TNF*, *TACE*, *TNFR1*, and *TNFR2* expression profiles within a healthy tissue and TCGA pan-cancer LUAD transcriptomic data set. We found a significant reduction of *TNF*, *TACE*, and *TNFR2* transcripts within the LUAD cohort (Figure 2A,B), leading to an increased *TNFR1*/*TNFR2* ratio (Figure 2C). These findings imply that LUAD tumors are characterized by reduced numbers of TNFα and especially of sTNFα molecules as TACE is significantly reduced. The specific reduction of TNFR2 and not TNFR1 in LUAD tumors further implies that TNFR1 but not TNFR2 plays a crucial role in tumor progression. As TNFR1 also shows high affinity for LTα3, we used the same LUAD cohort to evaluate the expression profile of the lymphotoxin alpha monomer encoding gene *LT**A*. We found a significant upregulation of *LTA* in the LUAD cohort compared to healthy controls (Figure 2D). Previous studies linked the expression of at least two LTA-signaling members to a poor clinical outcome in lung cancer patients: the receptor for the heterotrimeric LTαβ (LTβR) and the alternative LTα3 receptor herpes virus entry mediator (HVEM) [98,99]. Overall, these findings suggest that the TNFα pathway will have the highest likelihood to signal within the LUAD TME via TNFR1 through low-affinity binding to tmTNFα or high-affinity binding to LTα3, as graphically visualized with green arrows in Figure 3.

Nevertheless, studies on the lung tumor growth promoting role of TNFα signaling are ample, too. First, several clinical studies have reported on the decisive role of TNFα in lung cancer EMT, invasion, and metastasis [101,102]. Secondly, TNFα has been shown to play an essential role in creating an immunosuppressive lung TME [103], among others, by upregulating MHC-II in alveolar type-II (AT-II) cells responsible for a plethora of functions that support the maintenance and optimal functioning of alveoli. Because AT-II cells’ primary function is not antigen presentation, they lack co-stimulatory signals (CD80/CD86), essential to effectively prime CD4^+^ T cells. Thus, instead of creating an antitumor immune response, increased AT-II cell-specific MHC-II expression can trigger Treg differentiation [104]. Numerous studies evaluated TNFR2 as a potential biomarker for NSCLC as it has been shown to be crucial for TNFα-mediated immunosuppression [105,106]. High amounts of TNFR2^+^ Tregs have been found in the TME of murine [39,107] as well as human advanced lung cancers [108,109], suggesting that Tregs are activated through TNFα/TNFR2 signaling. In addition, TNFR2 expression by lung tumor cells and the lung TME has been shown to support tumor cell survival [110], pre- metastatic niche formation [111], and neovascularization via vascular endothelial growth factor (VEGF) release. These findings are further supported by the observation that the levels of TNFR2 expression in human lung cancer patients, which are up to 35% [106], correlate positively to a more advanced clinical stage, immune invasion, progressive metastasis, shorter survival time, and poor prognosis [106,108]. These findings were confirmed in TNFR2-ko mice engrafted with the Lewis lung carcinoma (LLC) model. Compared to wild type and TNFR1/2 double ko mice, tumor growth decreased twofold in TNFR2-ko mice specifically and correlated with reduced VEGF expression and capillary density, along with increased numbers of apoptotic LLC cells. As they further showed that blocking TNFR2 via a short-hairpin RNA in cultured LLCs increased TNFα-mediated apoptosis and expression of several angiogenic factors, they confirmed that TNFR2 directly ameliorates angiogenesis and LLC survival [110]. Moreover, Chopra et al. reported that TNFα or TNFR2 deficiency on immune cells resulted in the reduction of lung metastasis and a decrease in the number of pulmonary Tregs [39].

In conclusion, preclinical evidence suggests that TNFR2 is involved in lung tumor progression while tmTNFα (with high affinity for TNFR2) has been linked to a better prognosis for lung cancer patients. This implies that the biomarker potential of tmTNFα and TNFR2 for lung cancer progression most likely pivots on the delicate balances of sTNFα over tmTNFα as well as TNFR1 over TNFR2. 

## 4. Linking TNFα to Antitumor Immunotherapy in Lung Cancer

The goal of antitumor immunotherapy is to completely and specifically eradicate both primary and metastatic tumor lesions by mobilized cytotoxic effector cells. Hence, immunotherapy can achieve actual cures of advanced lung cancer patients, representing an unprecedented reality [112,113]. Therefore, the first FDA approval of an immunotherapeutic treatment for squamous cell NSCLC benchmarked a revolutionary era for lung cancer patients. This treatment is based on blocking the immune checkpoint programmed death-1 (PD-1) pathway. Under healthy conditions this pathway is used to put an adequate brake on T cell stimulation and return to homeostatic conditions. As tumor cells can express the PD-1 ligand (PD-L1) themselves, they can corrupt this pathway to hinder their execution by PD-1^+^ TAA-specific cytotoxic effector cells [114,115]. Since 2016, five PD-(L)1 inhibitors (nivolumab, pembrolizumab, atezolizumab, durvalumab, and cemiplimab) have been approved by the FDA as second- and/or first-line treatment options for advanced NSCLC [93]. Notably, for the treatment of SCLC, both nivolumab and pembrolizumab were originally approved [116] yet have been withdrawn from the US market since confirmatory trials failed to evidence improved survival outcomes. Additionally, only ~20% of unselected NSCLC patients benefit from blocking immune checkpoints, and many of the initial responders eventually develop resistance to therapy. Moreover, the growing trend to combine several immune checkpoint inhibitors (ICIs) coincides with a growing occurrence of severe to fatal immune-related adverse effects (irAEs), often related to a local increase in TNFα [117]. Together with the emerging concept of hyperprogression upon ICI [118], these phenomena cast light on the current knowledge gap of immunotherapy hampering mechanisms. 

In search for clues, the relationship between TNFα and immune checkpoint signaling in the TME is being explored, hinting towards a lead role for TAMs. While IFN-γ is the main regulator of PD-L1 expression in tumor cells, PD-L1 expression in TAMs seems to be regulated via TNFα [119]. In 2017, Hartley et al. demonstrated that TNFα increases the expression of PD-L1 on bone marrow-derived monocytes and macrophages. They found that this was maintained through the secretion of versican by tumor cells, which stimulated the production of TNFα by monocytes themselves in a TLR2-dependent manner [120]. One year later, the same group provided more evidence on the interactions between the TNFα and PD-L1 pathways, as they demonstrated that PD-L1 blockade increased spontaneous macrophage proliferation, survival, and activation in vitro. Via RNAseq and IPA software analysis of these anti-PD-L1 treated macrophages, they further revealed an activated TNFR2 signaling profile [119]. Furthermore, it was recently shown in NSCLC patients that TNFα-secreting TAMs can enhance hypoxia and aerobic glycolysis and that TAMs dampen PD-L1 expression on murine lung tumor cells specifically [121,122]. The latter does contradict the observation that TAM-secreted TNFα could stabilize PD-L1 expression on 4T1 mammary cancer cells, triggering immunosuppression in vivo [123]. 

As melanoma remains the textbook example for immunotherapy responsiveness, TNFα targeting studies are most numerous for this cancer type. Overall, preclinical TNFα blockade has been shown to reduce the induction of irAEs upon ICI combinations and even improve therapeutic effectiveness of ICIs [117,124]. Upon adoptive CD8^+^ T cell transfer, TNFα appeared to be a crucial factor in the incitement of melanoma dedifferentiation, which resulted in immune escape and melanoma relapse [125]. Bertrand et al. partly explained these effects by the observation that TNFα/TNFR1 signaling triggers AICD of tumor-infiltrating CD8^+^ T cells in melanoma, with subsequent lack of response to anti-PD-1 therapy [126]. Hence, via systemic administration of etanercept, melanoma growth was inhibited in immunocompetent animals specifically. Notably, similar effects were seen in TNFR1-ko, but not TNFR2-ko, mice, suggestive for the decisive role of TNFR1 in this AICD of CD8^+^ T cells [127]. A few years later, Bertrand et al. further validated these findings by showing that anti-PD-1 therapy can stimulate T-cell expression of the alternative checkpoint T-cell immunoglobulin and mucin domain 3 (TIM-3) via TNFα. Moreover, they could demonstrate that co-blockade of PD-1 and TNFα overcomes resistance to anti-PD-1 monotherapy [124]. Hence, we eagerly await the results from the first Phase Ib, open-label trial [128] that is evaluating the administration of nivolumab (anti-PD-1) and ipilimumab (anti-CTLA-4) in combination with the anti-TNFα drug infliximab or certolizumab in patients with advanced melanoma. 

In contrast to melanoma, only a handful of preclinical studies have explored the potential of TNFα pathway modulating strategies to treat lung cancer with rather contradictory findings. For example, anti-TNFα antibody treatment has been tested together with an intrapulmonary IFN-beta immuno-gene therapy (Ad.IFNβ) in an orthotopic mouse model of lung cancer. The rationale for this was that blocking the pro-inflammatory actions of TNFα could reduce the induction of dose-limiting pulmonary inflammation upon Ad.IFNβ delivery. However, upon administration, the anti-TNFα antibody not only significantly reduced the pulmonary inflammation but also the therapeutic effect of Ad.IFNβ delivery. Hence, TNFα proved to be both a dose-limiting factor as well as crucial for the anti-tumor immune stimulatory capacity of Ad.IFNβ [129]. More straightforward was the negative role described for TNFα secretion by TAMs, as this promoted cell glycolysis, tumor hypoxia, and decreased PD-L1 expression. Hence, when the TNFα-secreting TAMs were depleted upon clodronate treatment of LLC-engrafted mice, PD-L1 expression significantly increased in the aerobic cancer cells. Moreover, this treatment increased tumor T cell infiltration and most importantly, its response to anti-PD-L1 therapy, which was otherwise completely ineffective [121]. While the above study suggests that TNFα blockade can increase effectivity of anti-PD-L1 therapy, this seems diametrically opposed to the observation that not reduced but increased serological levels of TNFα were found to be associated with improved anti-PD-1 treatment response and survival in NSCLC (along with IFN-γ, IL-1, IL-2, IL-4, IL-5, IL-6, IL-8, IL-10, and IL-12) [130]. Using an online available RNAseq dataset on CD8^+^ T cells sorted out of NSCLC patients’ peripheral blood mononuclear cells before and during anti-PD-1 mAb therapy [131], we were able to support the notion that increased levels of sTNFα could be linked to improved anti-PD-1 treatment response, as we found a non-significant increase in *TACE* expression within the responder group specifically, linked to more release of sTNFα from its tmTNFα form (Figure 2E). While the *TNFR1*/*TNFR2* ratio decreased upon anti-PD-1 treatment, this phenomenon was seen in both the responder and non-responder groups (Figure 2F). Finally, the levels of *LTA* were very similar between the responder and non-responder groups (data not shown), although we were unable to draw any conclusions for *TNF* due to the lack of representative data, underscoring the current lack of human data on gene expression profiles of the different TNFα family members before, during, and after antitumor immunotherapy.

## 5. Conclusions and Future Perspectives on TNFα Modulation for Lung Cancer Treatment

The multitude of contradictory findings currently poses a stalemate for TNFα pathway-affecting strategies in combination with immunotherapy to treat lung cancer and suggests the need for additional research into biomarkers to guide rationalized therapy combinations. This conundrum is reflected by the range of preclinical studies that report on the therapeutic efficacy of TNFα upon its administration as well as its inhibition [132]. To illustrate, when a TNF-based Activity-on-Target cytokine (AcTakine) was specifically targeted to the CD13^+^ neovasculature in vivo, the rapid destruction of the tumor neovasculature and complete regression of large, established tumors was demonstrated. In contrast, selective blockage of sTNFα via INB03 led to a reduced carcinogen-induced tumor incidence and growth rate [133]. Moreover, a detrimental role has been attributed to sTNFα and TNFR1 for melanoma-infiltrated functional CD8^+^ T cells as well as the onset of irAEs, rationalizing combined TNFα-blockade with immunotherapy to treat melanoma. 

By mining existing next-generation sequencing data from LUAD patients, the latter were shown to contain less *TNF*, and because of the significant reduction in *TACE*, sTNFα protein is likely to be most decreased. Together with the notion that tmTNFα, and not sTNFα, has been shown to play a key role in Th1-polarized antitumor immunity and improved lung cancer patient survival [18,134], this argues against a tumor promoting role for TNFα in lung cancer, discouraging TNFα-blockage for lung cancer treatment today. Additionally, the role of TNFR2 in lung cancer progression remains undetermined and requires more research. High amounts of TNFR2^+^ Tregs have been found in the TME of human advanced lung cancers, and TNFR2 has recently been identified as a tumor-promoting oncogene with new biomarker potential for cancer [135,136]. However, upon mining the currently available transcriptomic dataset from a TCGA LUAD patient cohort, we demonstrated that the expression of *TNFR2* is markedly decreased in the lung TME. Moreover, pre-clinically, TNFR2 agonists as well as antagonists have been linked to antitumoral effects, arguing against the effectiveness of TNFR2 modulation for lung cancer therapy [137,138,139,140]. 

To conclude, numerous studies point out that TNFα signaling is extensively involved in lung tumor progression and response to (immuno-)therapy. However, the underlying mechanisms of the different TNFα family members that modulate tumor prognosis and response to treatment remain to be revealed. We summarized our main findings in Figure 3 to highlight that TNFα signaling involves different ligands and receptors in LUAD, which have been linked to prognosis response to PD-(L)1 treatment. To make matters more complicated, all of these components can be expressed, secreted, and sensed by a broad range of malignant, immune, and non-immune cells within the respiratory tract, whereas most (pre-)clinical data have been based on serological values of sTNFα and RNA sequencing data. Therefore, we believe that larger genomic, transcriptomic, and proteomic dataset analysis studies are needed for various disease stages and treatment options on the single cell lung TME level to advance our current understanding of the biomarker and modulatory potential of the TNFα pathway for lung cancer prognosis and therapy.

## Figures and Tables

**Figure 1 ijms-22-08691-f001:**
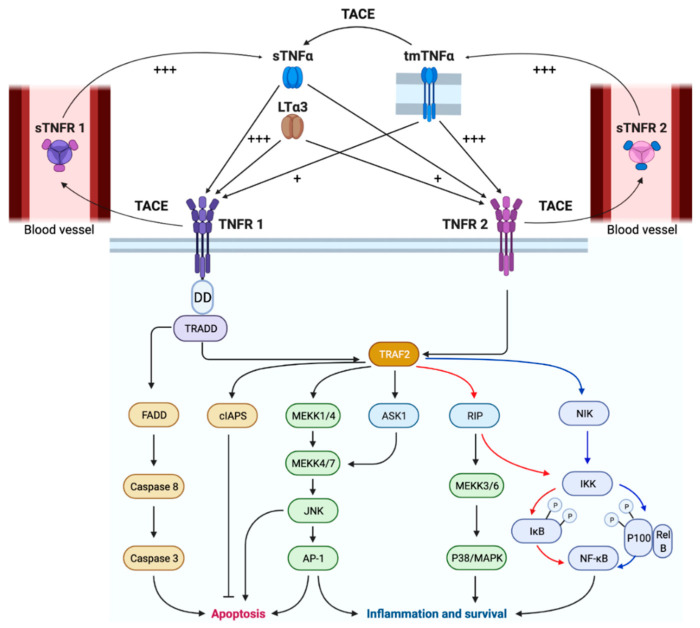
Pathways activated by soluble TNFα (sTNFα) and transmembrane TNFα (tmTNFα) upon binding to TNFRs. Downstream signaling via TNFR1 is most effective upon high-affinity (+++) binding to sTNFα. In the case of TNFR1, this can lead to caspase8/3-mediated apoptosis through signaling via its death domain (DD), recruitment of TRADD, and subsequent recruitment of FADD or TRAF2. The latter could also result in activation of pro-inflammatory signals via the "classical NF-κB" pathway, and is primarily activated via TNFR1 (red arrows). Upon high-affinity (+++) binding of tmTNFα to TNFR2, TRAF2 is triggered, which preferentially results in NF-κB activation via the alternative pathway (blue arrows) to activate the expression of proliferation and survival related genes. Both sTNFα and tmTNFα bind with low-affinity (+) to TNFR2 and TNFR1 respectively. TNF-alpha-converting enzyme (TACE), responsible for the conversion of sTNFα from tmTNFα and of soluble TNFRs [56]. For completeness, the alternative TNFR ligand lymphotoxin alpha 3 (LTα3), with its affinity binding profile to TNFR1 and 2, is depicted as well. This figure was created with BioRender.com (accessed on 4 June 2021).

**Figure 2 ijms-22-08691-f002:**
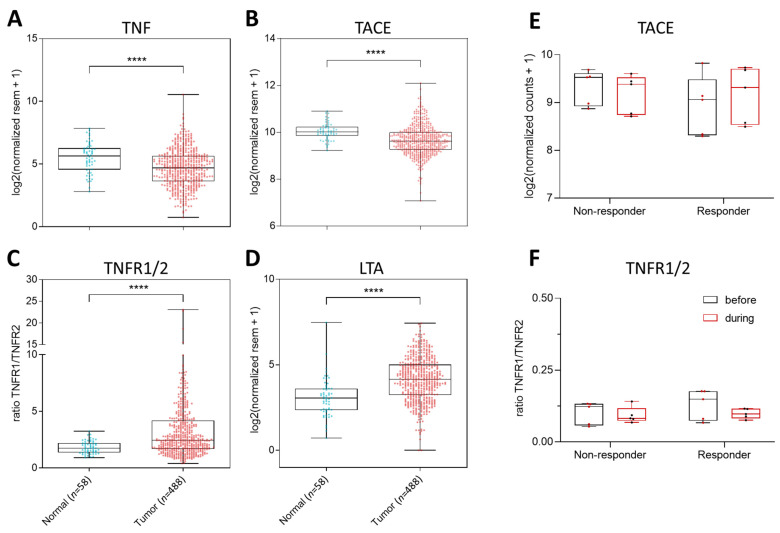
Gene expression analysis of several TNF family members within lung cancer patients. (**A**–**D**) Expression profiles were collected from TCGA for both normal (*n* = 58) and LUAD tumor tissue (*n* = 488). Next, we obtained the Log2-transformed normalized RSEM values via Wanderer for *TNF* (**A**), *TACE* (**B**), *TNFR1* and *2* (**C**), and *LTA* (**D**) [100]. For (**C**), the ratio of the expression values of *TNFR1* over *TNFR2* is depicted. Statistical analysis was performed using the Mann–Whitney test (****, *p* < 0.0001). (**E**,**F**) Gene expression analysis of *TACE*, *TNFR1*, and *TNFR2* in CD8^+^ T cells from NSCLC patients before and during anti-PD-1 mAb treatment. RNAseq dataset obtained from the Gene Expression Omnibus (GEO) database (accession number: GSE111414, accessed on 06/07/2021). Counts were normalized using DESeq2 in R. Counts for TNF were zero for most samples (data not shown). (**A**) Gene expression profile of *TACE* in both non-responder and responder groups. (**B**) Ratio of *TNFR1* and *TNFR2* normalized expression values. For each group: *n* = 5.

**Figure 3 ijms-22-08691-f003:**
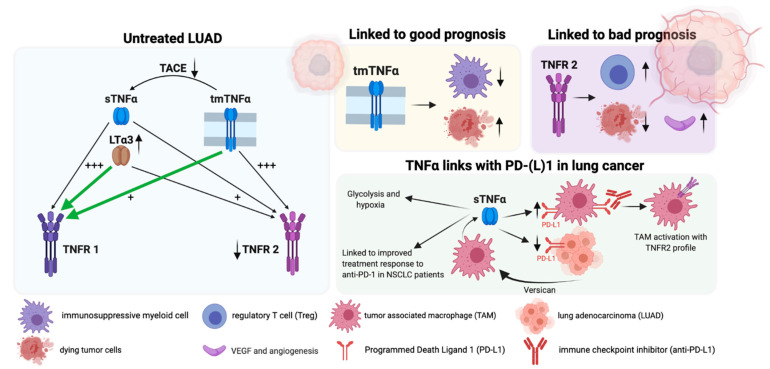
Overview of TNFα signaling within LUAD with link to prognosis and checkpoint inhibition. (+++) high-affinity binding, (+) low-affinity binding, green arrows support likeliness of TNFα-signaling pathways in LUAD patients. This figure was created with BioRender.com (accessed on 5 August 2021).

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
