# Peer review of "TNFα and Immune Checkpoint Inhibition: Friend or Foe for Lung Cancer?"

_ijms, 2021, doi:10.3390/ijms22168691_

Round 1

Reviewer 1 Report

Benoot et al., propose to review the scientific literature of TNF? and its clinical relevance for treatment of lung cancer during checkpoint inhibitor therapy. Despite very contradictory literature, authors have made an extensive effort to summarize and propose hypothesis to explain this data discrepancy. However, I would advise to consider some of the following raised points below to complete and improve this review.

Major points.

1/ Authors have tried to describe separately tmTNF vs. sTNF biological effect to decipher and interpret current literature. However, TNFR1 and TNFR2 also bind lymphotoxin α (LTα). Therefore, the review would greatly benefit to discuss LTα and interaction with ICI immunotherapy. For example, add a graph showing variation of LTα gene in responder vs. non responder could increase impact and message of this review. LTα and TNF interplay have been reviewed here: Cancers 2021, 13, 1775. https://doi.org/10.3390/cancers13081775.

2/ Authors should provide a graphical abstract summarizing putative action of TNF described in the review to help follow rational of the review. As it is now, readers struggle to draw a home-taking message.

Minor points.

1/ Quality of Figure 1 should be upgraded – Images looks blurry.

2/ For each histogram showing gene expression, please write gene presented directly on the figure for easier reading.

Reviewer 2 Report

The review “TNFa and immune checkpoint inhibition: friend or foe for lung cancer” by Benoot et al is taking up an important area of research where new treatment options for cancer are discussed. Although the review is interesting it does need some additional work.

Major comments:

My main concern is that some references seem not to be correctly cited as they do not describe the study stated in the review. Additionally, I think some of the reviews used as references can be replaced by original articles as that will be helpful to the reader.

Minor comments:

  1. Line 36 “…represented by a 55 and 75 kDa…..” are the references 4 and 5 correct for this statement? Please check.
  2. Line 49-51, is the reference 15 correct here? This reference seems to be about structural biology of the TNF family proteins.
  3. Line 56 to 59, “Hence, TNFa neutralizing antibodies have been shown……” the reference (16) is a review article. Although it is not wrong to use a review as a reference as more specific details are given in this sentence an original article for the results should be stated.
  4. Line 61-63, please check and rewrite this sentence.“To illustrate, when DCs express tmTNFa, this can interact with TNFR2……” the world (this) what is it refereeing to?
  5. Line 77-79, “TNFR1, also known as tumor………” the sentence states that TNFR1 is expressed on various tumor cell types, “ …….including various tumour cell types.” , however the reference (23) is a study of fibrosarcoma, so one tumour cell type. Please check this.
  6. Please check that reference 28 is correctly cited. Line 83-84.
  7. Line 99-101, “Upon an intracellular lack of cellular ……….” Please check this sentence as it seems that a part is missing.
  8. Line 108-110, “Additionally, both TNFRs can regulate gene expression by……” is reference 45 and 46 correct for P38?
  9. Figure 1, please state which software you have used to make the figure.
  10. Line 165-168, “ Specifically, chronic TNFa/TNFR1 binding increases…….” Is reference 65 correct for TAK-1? Please check.
  11. Line 189-190, “During the equilibrium phase……..” is reference (5) correct here? Please check.
  12. Line 208-212, “To illustrate, a marked increase in ……….” Please put a reference for this sentence.
  13. Line 233-235, “Lung cancer withholds two key…..” firstly please change the spealing of the (withold) to (withhold), secondly please put in a reference for this sentence.
  14. Line 302-304, “Moreover, Chopra et al, reported that ………” please add a reference for this sentence, specially if you are writing (Chopra et al).
  15. In your reference list, some studies seem to be there twice. Reference 25 and 43 are the same article, please check. Also reference 91 and 95 are the same.
  16. Some references have the first world of the title at the end, please see reference 40 or 60.
  17. Please give a link to your reference 77.

Round 2

Reviewer 1 Report

Authors have satisfied my expectations by updating their figure and text with LTa pathways and providing a graphical summary which improves greatly the scientific message.